# Prognostic Value of [^18^F]FDG PET Radiomics to Detect Peritoneal and Distant Metastases in Locally Advanced Gastric Cancer—A Side Study of the Prospective Multicentre PLASTIC Study

**DOI:** 10.3390/cancers15112874

**Published:** 2023-05-23

**Authors:** Lieke C. E. Pullen, Wyanne A. Noortman, Lianne Triemstra, Cas de Jongh, Fenna J. Rademaker, Romy Spijkerman, Gijsbert M. Kalisvaart, Emma C. Gertsen, Lioe-Fee de Geus-Oei, Nelleke Tolboom, Wobbe O. de Steur, Maura Dantuma, Riemer H. J. A. Slart, Richard van Hillegersberg, Peter D. Siersema, Jelle P. Ruurda, Floris H. P. van Velden, Erik Vegt

**Affiliations:** 1Biomedical Photonic Imaging Group, University of Twente, 7522 NB Enschede, The Netherlands; 2Department of Radiology, Leiden University Medical Center, 2333 ZD Leiden, The Netherlands; 3Department of Surgery, University Medical Center Utrecht, 3584 CX Utrecht, The Netherlands; 4TechMed Centre, University of Twente, 7522 NB Enschede, The Netherlands; 5Department of Radiology and Nuclear Medicine, University Medical Center Utrecht, 3584 CX Utrecht, The Netherlands; 6Department of Surgery, Leiden University Medical Center, 2333 ZD Leiden, The Netherlands; 7Multi-Modality Medical Imaging Group, TechMed Centre, University of Twente, 7522 NB Enschede, The Netherlands; 8Department of Nuclear Medicine and Molecular Imaging, University of Groningen, University Medical Center Groningen, 9713 GZ Groningen, The Netherlands; 9Department of Gastroenterology, Radboud University Medical Center, 6525 GA Nijmegen, The Netherlands; 10Department of Radiology and Nuclear Medicine, Erasmus MC University Medical Center Rotterdam, 3015 GD Rotterdam, The Netherlands

**Keywords:** [^18^F]FDG-PET/CT, gastric cancer, radiomics, machine learning

## Abstract

**Simple Summary:**

Patients with locally advanced gastric cancer have a five-year survival rate of 36–45% after curatively intended D2-gastrectomy combined with perioperative chemotherapy. This relatively poor survival is mainly due to recurrence of the disease. The aim of this study was to improve detection of peritoneal and distant metastases on [^18^F]FDG-PET images in patients with advanced gastric cancer using radiomics. Radiomics consists of the extraction of large amounts of quantitative features from medical imaging and the subsequent mining of this dataset for potential information to monitor disease characteristics in clinical practice. Three classification models were developed to determine the added value of radiomics: a model with clinical variables only, a model with radiomic features only, and a clinicoradiomic model, combining clinical variables and radiomic features. [^18^F]FDG-PET-based radiomics showed no additional value in predicting peritoneal and distant metastases in locally advanced gastric cancer patients.

**Abstract:**

Aim: To improve identification of peritoneal and distant metastases in locally advanced gastric cancer using [^18^F]FDG-PET radiomics. Methods: [^18^F]FDG-PET scans of 206 patients acquired in 16 different Dutch hospitals in the prospective multicentre PLASTIC-study were analysed. Tumours were delineated and 105 radiomic features were extracted. Three classification models were developed to identify peritoneal and distant metastases (incidence: 21%): a model with clinical variables, a model with radiomic features, and a clinicoradiomic model, combining clinical variables and radiomic features. A least absolute shrinkage and selection operator (LASSO) regression classifier was trained and evaluated in a 100-times repeated random split, stratified for the presence of peritoneal and distant metastases. To exclude features with high mutual correlations, redundancy filtering of the Pearson correlation matrix was performed (r = 0.9). Model performances were expressed by the area under the receiver operating characteristic curve (AUC). In addition, subgroup analyses based on Lauren classification were performed. Results: None of the models could identify metastases with low AUCs of 0.59, 0.51, and 0.56, for the clinical, radiomic, and clinicoradiomic model, respectively. Subgroup analysis of intestinal and mixed-type tumours resulted in low AUCs of 0.67 and 0.60 for the clinical and radiomic models, and a moderate AUC of 0.71 in the clinicoradiomic model. Subgroup analysis of diffuse-type tumours did not improve the classification performance. Conclusion: Overall, [^18^F]FDG-PET-based radiomics did not contribute to the preoperative identification of peritoneal and distant metastases in patients with locally advanced gastric carcinoma. In intestinal and mixed-type tumours, the classification performance of the clinical model slightly improved with the addition of radiomic features, but this slight improvement does not outweigh the laborious radiomic analysis.

## 1. Introduction

Gastric cancer is the third most common cause of cancer-related death worldwide [1]. The prognosis after curatively intended treatment is relatively poor, with a five-year survival rate of 36–45% after a D2-gastrectomy combined with perioperative chemotherapy. The main reasons for failure of curative treatment are the detection of distant metastases during neoadjuvant chemotherapy, unexpected intraoperative peritoneal metastases, or tumour irresectability at the onset of gastrectomy, or local or distant recurrences shortly after surgery [2,3]. In the Netherlands, only 60% of gastric cancer patients undergo curative D2-gastrectomy, since the remaining patients present with irresectable tumours or distant metastases [4]. After detecting distant metastases or irresectable disease, treatment is changed from curative to palliative intent. Hence, accurate primary staging is crucial.

Currently, computed tomography (CT) of the thorax and abdomen is performed to detect any metastases. However, the sensitivity of CT to detect peritoneal and distant metastases is low with 22–33% and 14–65%, respectively [5,6,7]. The Dutch multicentric PLASTIC study assessed the diagnostic performance and clinical and financial impact of 2-[^18^F]fluoro-2-deoxy-D-glucose positron emission tomography combined with CT ([^18^F]FDG-PET/CT) and staging laparoscopy in addition to initial staging with CT for locally advanced gastric cancer (cT3-4 and/or cN+) [8]. Nevertheless, the sensitivity of visual assessment of [^18^F]FDG-PET/CT for the detection of distant metastases was only 33% (95% CI: 17–53%), and the PLASTIC-study did not find additional value of qualitative assessment of [^18^F]FDG-PET/CT in gastric cancer [9]. However, medical images might contain more information than can be assessed visually. Therefore, quantitative assessment using radiomics might be of added value [10]. Radiomics, the extraction of large amounts of quantitative features from medical imaging and subsequent mining of this dataset for potential information useful for quantification or monitoring of tumour or disease characteristics in clinical practice, is a rapidly evolving field in medical imaging [11,12]. Radiomics aims to find stable and clinically relevant image-derived biomarkers that may provide new insights in tumour biology and guide patient management.

Several studies have shown promising results of CT-based radiomics for the identification of metastases in gastric cancer [13,14,15], but only a few studies investigated the predictive value of [^18^F]FDG-PET/CT radiomics [16,17]. Hence, the added value of [^18^F]FDG-PET/CT-radiomics for gastric cancer is unclear. 

The aim of this study was to assess the added value of [^18^F]FDG-PET-based radiomics and clinical characteristics of the primary tumour for the identification of peritoneal and distant metastases in patients with advanced gastric cancer. 

## 2. Materials and Methods

### 2.1. Patient Population

[^18^F]FDG PET scans of the multicentre PLASTIC study were analysed [9]. The PLASTIC study assessed the diagnostic performance and clinical and financial impact of [^18^F]FDG-PET/CT and staging laparoscopy in addition to initial staging with CT in patients with surgically resectable, locally advanced gastric adenocarcinoma (cT3-4b, N0-3, M0). After the initial CT, patients underwent an [^18^F]FDG-PET/CT, followed by a staging laparoscopy if [^18^F]FDG-PET/CT was found negative, as standard of care according to the Dutch national guidelines [4]. The presence of peritoneal and distant metastases was confirmed based on (histo)pathological biopsy/cytology and/or follow-up imaging. Because this study did not allocate patients to interventions other than standard of care according to national guidelines, the study did not fall within the Medical Research Involving Human Subjects Act (WMO). A non-WMO declaration (METC 16-633/C) had been obtained from the Medical Ethical Review Board of the University Medical Center Utrecht. In addition, the trial was approved by the institutional review boards in each of the 18 participating centres (Trial registration: NCT03208621. The PLASTIC-study was regis-tered prospectively on 30 June 2017).

### 2.2. Image Acquisition and Reconstruction

[^18^F]FDG-PET/CT acquisition was preferably performed following the European Association of Nuclear Medicine (EANM, Vienna, Austria) guidelines version 2.0 for tumour PET imaging [18]. PET/CT scanners were required to have EANM Research Ltd. (EARL) accreditation, but when EARL-compliant PET images were not available, the PET images were reconstructed according to the site-specific reconstruction protocol. Patients had to refrain from exercise and fast for at least 4 to 6 h before the injection of [^18^F]FDG. Patients were prehydrated by drinking approximately 1 litre of water in the 2 h before injection. Fasting blood glucose had to be preferably below 11 mmol/L. After the injection of [^18^F]FDG, patients remained seated or lying and silent for 1 h in a warm room. The acquisition of a PET scan from eyes to thighs started approximately 60 min (range 55–75 min) after the injection of [^18^F]FDG, being accompanied by a low-dose CT of the same scanning range [8]. 

### 2.3. Quantitative Image Analysis

#### 2.3.1. VOI Delineation

Volume of interest (VOI) delineation was performed using 3DSlicer version 4.11.2 (www.slicer.org, accessed on 1 June 2022) [19] and in-house built software implemented in Python version 3.7 (Python Software Foundation, Wilmington, DE, USA). Tumours were delineated on [^18^F]FDG PET scans using an isocontour that applies an adaptive threshold of 50% of the peak standardised uptake value (SUV_peak_), obtained using a sphere of 12 mm diameter [20], corrected for local background [21]. Boxing was applied to exclude surrounding [^18^F]FDG-avid tissues.

#### 2.3.2. Radiomic Feature Extraction

Radiomic feature extraction was performed in PyRadiomics version 3.0 in Python version 3.7 (Python Software Foundation, Wilmington, DE, USA) [22]. For each VOI, 105 radiomic features were extracted: 18 first order features, 14 shape features, and 73 texture features (22 grey level co-occurrence matrix (GLCM), 16 grey level run length matrix, 16 grey level size zone matrix, 14 grey level dependence matrix, 5 neighbouring grey tone difference matrix). In addition, the total lesion glycolysis, the product of the mean SUV and the metabolic tumour volume, was calculated. A fixed bin size of 0.5 g/mL was applied, and images were interpolated to isotropic voxels of 4.00 × 4.00 × 4.00 mm^3^ using B-spline interpolation, with grids aligned by the input origin.

### 2.4. Statistical Analysis

The statistical analysis was performed in Python version 3.7 (Python Software Foundation, Wilmington, DE, USA) and Orange Data Mining (University of Ljubljana, Ljubljana, Slovenia) [23]. A schematic overview of the analysis can be found in Figure 1.

ComBat harmonisation was used to harmonise the features extracted from the non-EARL-compliant images to features of the EARL-compliant-images using Python package neuroCombat version 0.2.10 [24,25]. Since raw imaging data were not stored, it was not possible to harmonize images posteriori. Therefore, other ways to harmonise radiomic features were investigated. ComBat harmonisation is a method that originates from genomic research and is able to compensate for batch effects introduced, e.g., by scanners or protocols. ComBat was directly applied to features already extracted from the images. A Kolmogorov–Smirnov test was performed to test differences in the distribution of SUV_mean_ between EARL-compliant and non-EARL-compliant images, and ComBat was applied in case of a significant difference in distributions [26].

A least absolute shrinkage and selection operator (LASSO) regression classifier was trained and evaluated in a 100-times repeated random split, stratified for the presence of peritoneal and distant metastases. In each split, 80% of the data were used for training of the model and 20% for testing. Radiomic features were standardised (µ = 0, σ = 1) to prevent a large contribution of high-valued features. To exclude features with high mutual correlations, redundancy filtering of the Pearson correlation matrix was performed (r = 0.9). The redundancy filtered feature matrix was used as an input for LASSO regression. Model performances are expressed by the area under the receiver operating characteristic curve (AUC) of the test sets. 

In addition to the radiomic model, a model based on the clinical characteristics age, sex, clinical T-stage, clinical N-stage, tumour location, Lauren classification, degree of differentiation, and Her2Neu status was built. The clinical variables Lauren classification, degree of differentiation, and Her2Neu-status had many missing values (21%, 43%, and 54%, respectively). Therefore, these values were imputed based on the variables PET positivity, positivity of diagnostic laparoscopy, primary tumour positivity on PET, lymph node positivity on PET, presence of fluid on diagnostic laparoscopy, curative or palliative treatment, gastric resection performed, curative treatment plan, type of resection, type of treatment, chemotherapy scheme, recurrence after six months, location of recurrence after six months, and survival status after six months. A clinicoradiomic model was created, combining all clinical variables and selected radiomic features.

Moreover, subgroup analyses based on the Lauren classification were performed, since intestinal-type and diffuse-type tumours show different metastatic patterns and [^18^F]FDG-uptake [27]. For intestinal and mixed-type tumours, as well as for diffuse-type tumours, a clinical, radiomic, and clinicoradiomic model was developed. 

The findings were validated in a sham experiment [28]. The outcome labels were randomly shuffled for 100 iterations, and mean AUCs were calculated. Randomisation of the outcome labels preserves the distributions and the multicollinearity of the radiomic features and the prevalence of the outcome, but it uncouples their potential relation. In the sham experiment, AUCs of 0.50 were expected.

## 3. Results

A total of 236 patients were considered for radiomic analysis, of whom thirty were excluded. Reasons for exclusion were lesion could not be identified on [^18^F]FDG-PET/CT (*n* = 12), corrupt DICOM files (*n* = 10), or missing clinical variables (*n* = 8). Thus, 206 remaining patients with advanced gastric adenocarcinoma were analysed, of which 43 (21%) had metastases (Table 1, Figure 2).

SUV_mean_ values were significantly different between EARL-compliant and non-EARL-compliant images (p=0.04). Therefore, ComBat harmonisation was performed, which lead to no significant differences in SUV_mean_ (p=0.95). Therefore, results are presented for data harmonised using ComBat.

None of the models could identify metastases with mean AUCs in the test sets averaged for the 100 splits of 0.59, 0.51, and 0.56, for the clinical, radiomic, and clinicoradiomic model, respectively (Table 2, Figure 3). In the sham experiment, no model yielded a test AUC different from 0.50 (range: 0.49–0.51). 

Subgroup analysis based on the Lauren classification did slightly improve the classification performance for the intestinal and mixed-type tumours (Table 3, Figure 3). While the clinical model and the radiomic model alone both showed poor performances with AUCs of 0.67 and 0.60, respectively, the combination of clinical and radiomic features resulted in a moderate AUC of 0.71. In the sham experiment, no model yielded a test AUC different from 0.50 (range: 0.49–0.51).

Finally, subgroup analysis of diffuse-type tumours did not improve the classification performance with AUCs of 0.58, 0.53, and 0.56 for the clinical, radiomic, and clinicoradiomic model, respectively (Table 4, Figure 3). In the sham experiment, no model yielded a test AUC different from 0.50 (range: 0.49–0.51). 

## 4. Discussion

In this side study of the PLASTIC-study, we have built an [^18^F]FDG-PET radiomics model with the aim to preoperatively identify peritoneal and distant metastases in 206 patients with surgically resectable, advanced gastric adenocarcinoma (cT3-4b, N0-3, M0). We found that neither the radiomics model nor the clinicoradiomic model showed any added value in the identification of peritoneal and distant metastases. Subgroup analysis based on the Lauren classification, slightly improved the classification performance in intestinal and mixed-type tumours. Separately, the clinical model and the radiomic model showed poor AUCs, but the clinicoradiomic model resulted in a borderline moderate AUC. However, weighing the slight improvement in model performances against the laborious radiomic analysis, the additional value of radiomics was still limited for this subgroup, and therefore not clinically relevant. Moreover, diffuse-type tumours did not benefit from [^18^F]FDG-PET radiomics. Similarly to the original PLASTIC-study, which did not find an additional value of qualitative assessment of [^18^F]FDG-PET/CT in gastric cancer [9], this study also shows that quantitative assessment does not provide additional value. 

Several studies have investigated both CT and [^18^F]FDG-PET radiomics for the identification of metastases in gastric cancer, showing promising results [13,14,15,16,17]. However, to the best of our knowledge, only one study developed an [^18^F]FDG-PET radiomic model for the identification of peritoneal metastases in primary gastric cancer [17]. Similar to our study, this study by Xue et al. compared a clinical model, an [^18^F]FDG-PET-based radiomic model, and a clinicoradiomic model to identify peritoneal metastases in gastric cancer, resulting in validation AUCs of 0.87, 0.69, and 0.90, respectively. It is challenging to directly compare these results to ours, since Xue et al. only identified peritoneal metastases, while our work tried to identify peritoneal as well as distant metastases. However, a trend can be observed when comparing the results of Xue et al. to our subgroup analysis of the intestinal and mixed-type tumours. For both analyses, the clinical models performed better than the radiomic models and the clinical models were only slightly improved by the addition of radiomic features in the clinicoradiomic model. Therefore, the value of [^18^F]FDG-PET radiomics in both studies is limited when these slight improvements in model performances are compared to the laborious radiomic analysis. In addition, our clinical and clinicoradiomic models performed substantially worse than the models of Xue et al. This might have been caused by the imputation of our clinical variables. For three of the eight clinical variables, more than 20% of the values were missing. By excluding patients with missing data, almost 50% of the data would have been discarded. Therefore, missing variables were imputed based on fourteen other clinical and treatment variables. The imputation may have negatively impacted our clinical and clinicoradiomic models. 

Furthermore, a class imbalance might have complicated the classification task. Our dataset consisted of 21% of patients with peritoneal and distant metastases, compared to 31% of patients with peritoneal metastases in the study of Xue et al. As a result, the machine learning algorithm was trained using data with an overrepresentation of the characteristics of the majority class, while underrepresenting the characteristics of the minority class, i.e., the presence of metastases. To validate our findings, these AUCs were compared to AUCs of sham experiments, where the outcome labels were randomly shuffled. 

Subgroup analyses based on the Lauren classification were performed since intestinal-type and diffuse-type tumours show different metastatic patterns and [^18^F]FDG-uptake [27]. Diffuse-type tumours frequently present with peritoneal metastases, while intestinal-type tumours more often show other type of distant metastases, e.g., in the liver or lungs. Moreover, intestinal-type tumours show significantly higher [^18^F]FDG-uptake compared to diffuse-type tumours [27]. Furthermore, [^18^F]FDG-PET/CT has shown a higher sensitivity for detecting recurrence in gastric cancer in [^18^F]FDG-avid primary tumours compared to non-[^18^F]FDG-avid tumours [29]. It was hypothesised that radiomics would perform better in [^18^F]FDG-avid tumours (such as intestinal-type tumours) than in non-[^18^F]FDG-avid tumours (such as diffuse type tumours) since, in case of a fixed bin width, the larger range of voxel values within the VOI enables more variation in the values of some texture features. Ultimately, in intestinal and mixed-type tumours, a limited added value of radiomics was observed. 

Our study has several strengths and limitations. A strength is the relatively large number of patients. Most radiomic studies in nuclear medicine consider relatively small patient cohorts, as a result of the specialised nature of the imaging procedures compared to for instance CT. In addition, [^18^F]FDG-PET/CT scans were collected from sixteen health care institutes in the Netherlands, which is both a strength and a limitation. As a result of the multicentre setting, a larger cohort of patients could be obtained. In addition, [^18^F]FDG-PET/CT-scans were acquired using different scanners and reconstruction protocols, which is representative for the clinical practice. However, the variation in imaging protocols was also challenging, since not all scans were EARL-compliant. This increased variability and reduced the repeatability and reproducibility of the extracted radiomic features [30]. To minimise the difference between EARL-compliant images and images reconstructed with clinical, site-specific reconstruction protocols, ComBat harmonisation towards the EARL-compliant scans has been performed [26]. Another limitation of the study was the VOI delineation. Twelve patients were excluded because lesions could not be detected on [^18^F]FDG PET/CT. Low [^18^F]FDG-avidity also complicated the VOI delineation of some included lesions.

Although [^18^F]FDG-PET radiomics analysis was ineffective for the detection of peritoneal and distant metastases in gastric cancer, radiomics derived from other imaging modalities might be beneficial. Radiomics derived from contrast-enhanced CT has already shown promising results [13,14,15]. Since diagnostic CT is incorporated in the Dutch national guidelines of gastric carcinomas, future studies might focus on handcrafted or deep learning radiomic analysis of these scans. Recently, some studies suggested a potential added value of diffusion weighted magnetic resonance imaging and fibroblast-activation-protein-inhibitor (FAPI) PET for preoperative staging of gastric cancer [31,32]. Radiomics derived from these modalities may provide new insights in the tumour biology of gastric cancer.

## 5. Conclusions

Similarly to qualitative assessment of [^18^F]FDG PET, quantitative assessment using radiomics did not contribute to the preoperative identification of peritoneal and distant metastases in patients with surgically resectable, locally advanced gastric adenocarcinoma (cT3-4b, N0-3, M0) in a large Dutch multicentric patient cohort. In intestinal and mixed-type tumours, the classification performance of the clinical model slightly improves with the addition of radiomic features, but this slight improvement does not outweigh the laborious radiomic analysis.

## Figures and Tables

**Figure 1 cancers-15-02874-f001:**
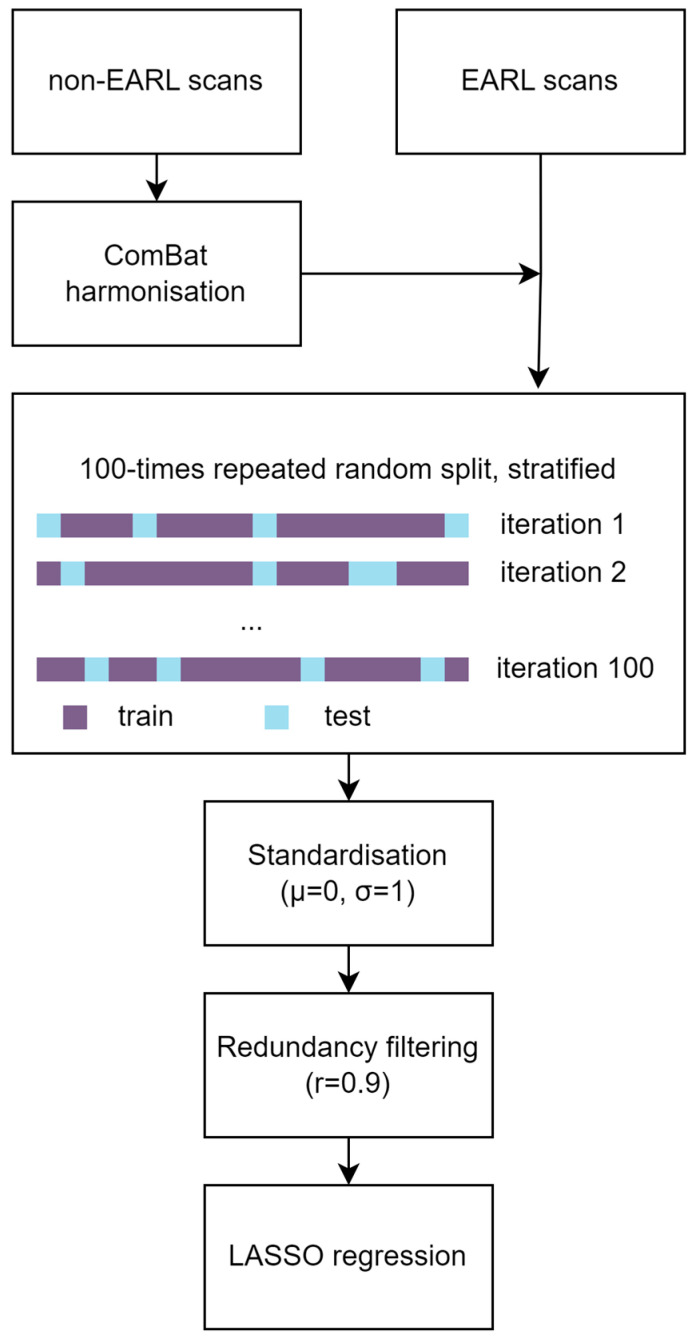
Schematic overview of the statistical analysis including ComBat harmonisation of the features from non-EARL-compliant images to EARL-compliant-images, 100-times repeated random split, stratified for the presence of metastases (80% training, 20% test), standardisation (µ = 0, σ = 1), redundancy filtering of the Pearson correlation matrix (r = 0.9), and fitting of a LASSO regression classifier.

**Figure 2 cancers-15-02874-f002:**
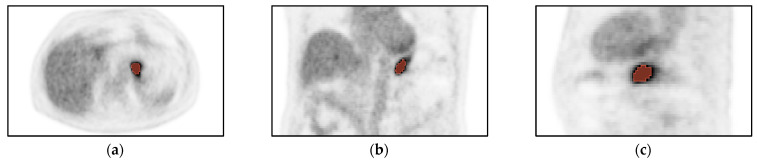
Volume of interest (VOI) of a tumour in the (**a**) transversal, (**b**) coronal, and (**c**) sagittal direction. Volumes of interest are indicated in orange.

**Figure 3 cancers-15-02874-f003:**
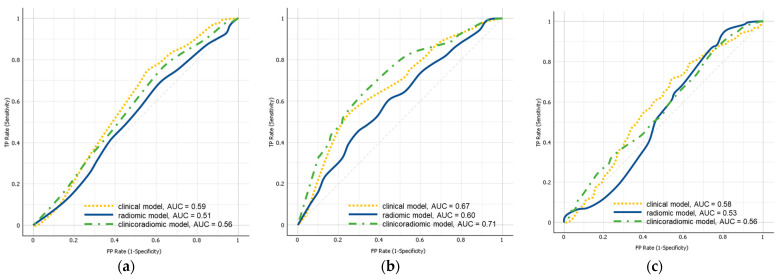
Mean test set AUCs averaged for the 100 splits of the clinical model (yellow; dashed), the radiomic model (blue; solid), and the clinicoradiomic model (green; dashed) for all included patients (**a**), and subgroups analyses for intestinal and mixed-type tumours (**b**) and diffuse-type tumours (**c**). AUC: area under the receiver operating characteristic curve.

**Table 1 cancers-15-02874-t001:** Clinical characteristics and traditional quantitative imaging parameters of included patients.

Characteristic (*n* (%))	Value
Age (years), median (range)	70 (35–87)
Sex	
Male	130 (63%)
Female	76 (37%)
Presence of metastases	
Yes	43 (21%)
No	163 (79%)
Clinical T-stage	
T3	15 (7%)
T4a	156 (76%)
T4b	33 (16%)
Missing	2 (1%)
Clinical N-stage	
N0	96 (47%)
N+	106 (51%)
Missing	4 (2%)
Tumour location	
Cardia	40 (19%)
Corpus & fundus	57 (28%)
Antrum & pylorus	85 (41%)
Diffuse	19 (9%)
Missing	5 (3%)
Lauren classification	
Intestinal and mixed	125 (61%)
Diffuse	81 (39%)
Differentiation	
Well	9 (4%)
Moderate	87 (42%)
Poor	107 (52%)
Undifferentiated	3 (2%)
Her2Neu status	
Positive	13 (6%)
Negative	193 (94%)
EARL-compliant PET scan	
Yes	94 (46%)
No	112 (54%)
SUV_max_ (g/mL), median (range)	6.9 (1.5–51.4)
MTV (cm^3^), median (range)	17.8 (2.6–135.0)

SUV_max_: maximum standardised uptake value, MTV: metabolic tumour volume.

**Table 2 cancers-15-02874-t002:** Mean test set AUCs of the clinical model, the radiomic model, and the clinicoradiomic model, averaged for the 100 splits. For all models, the included variables are specified; feature classes are in brackets.

Model	Variables	AUC
Clinical model	Age	0.59
	Sex	
	Clinical T-stage	
	Clinical N-stage	
	Tumour Location	
	Lauren classification	
	Differentiation	
	Her2Neu status	
Radiomic model	Small area low grey level emphasis (GLSZM)	0.51
	Grey level non-uniformity (GLRLM)	
	Inverse difference moment normalised (GLCM)	
	Grey level non-uniformity (GLSZM)	
	Small area emphasis (GLSZM)	
	Cluster prominence (GLCM)	
	Cluster shade (GLCM)	
	Large dependence high grey level emphasis (GLDM)	
	Size zone non-uniformity (GLSZM)	
	Sphericity (shape)	
	Elongation (shape)	
Clinicoradiomic model	All variables specified above	0.56

AUC: area under the receiver operating characteristic curve; GLCM: grey level co-occurrence matrix, GLRLM: grey level run length matrix; GLSZM: grey level size zone matrix; GLDM: grey level dependence matrix.

**Table 3 cancers-15-02874-t003:** Mean test set AUCs of the of clinical model, the radiomic model, and the clinicoradiomic model for the subgroup analysis of intestinal and mixed-type tumours, averaged for the 100 splits. For all models, the included variables are specified; feature classes are in brackets.

Model	Variables	AUC
Clinical model	Age	0.67
	Sex	
	Clinical T-stage	
	Clinical N-stage	
	Tumour Location	
	Differentiation	
	Her2Neu status	
Radiomic model	Skewness (shape)	0.60
	Correlation (GLCM)	
	Inverse difference moment normalised (GLCM)	
	Grey level non-uniformity (GLSZM)	
	Cluster prominence (GLCM)	
	Elongation (shape)	
Clinicoradiomic model	All variables specified above	0.71

AUC: area under the receiver operating characteristic curve; GLCM: grey level co-occurrence matrix; GLSZM: grey level size zone matrix.

**Table 4 cancers-15-02874-t004:** Mean test set AUCs of the clinical model, the radiomic model, and the clinicoradiomic model for the subgroup analysis of diffuse-type tumours, averaged for the 100 splits. For all models, the included variables are specified; feature classes are in brackets.

Model	Variables	AUC
Clinical model	Age	0.58
	Sex	
	Clinical T-stage	
	Clinical N-stage	
	Tumour Location	
	Differentiation	
	Her2Neu status	
Radiomic model	Contrast (NGTDM)	0.53
	Strength (NGTDM)	
	Correlation (GLCM)	
	Elongation (shape)	
	Small area low grey level emphasis (GLSZM)	
	Skewness (first order)	
Clinicoradiomic model	All variables specified above	0.56

AUC: area under the receiver operating characteristic curve; GLCM: grey level co-occurrence matrix; GLSZM: grey level size zone matrix; NGTDM: neighbouring grey tone difference matrix.

## Data Availability

The datasets generated during and/or analysed during the current study are available from the corresponding author upon reasonable request.

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
