# Peer review of "Prognostic Value of [18F]FDG PET Radiomics to Detect Peritoneal and Distant Metastases in Locally Advanced Gastric Cancer—A Side Study of the Prospective Multicentre PLASTIC Study"

_cancers, 2023, doi:10.3390/cancers15112874_

Round 1

Reviewer 1 Report

The presented study aims to assess the prognostic value of 18F-FDG imaging on gastric cancer patients using a retrospective radiomics approach. This study is a side project to a nationwide multi-centered PLASTIC clinical study. The authors extracted over 100 features out for an overall comparison; however, none of the features showed significance when tested with the outcomes of patients. The conclusion is straightforward, following the results that the use of FDG PET imaging has a limited improvement of prognosis over radiomics analysis. 

Despite the overwhelming and clearly stated format, some critical questions remain unsolved in this study. 

1) It is well-known that SPECT and PET are limited in their spatial resolution and anatomic information. The attenuation was calibrated by the CT while the image reconstruction to eliminate the noises. I don't find a clear statement whether the authors did radiomics analysis using both CT and PET information or, basically, PET. The only figure (Fig 1) suggested the VOI of the PET image, so it is anticipated that the radiomics was done using the PET signals only. Compared to CT, PET radiomics are relatively rare. The authors may want to have a more comprehensive discussion or, alternatively, include CT radiomics to suggest if PET or CT images are more feasible for radiomics approaches. 

2) Challenges for PET radiomics studies have been widely discussed in many articles. The authors may want to address or comment on the existing arguments, especially when their findings were negative to the use of radiomics prognosis. 

3) It is suggested to include an overview or flow chart for the radiomics analysis using PET data. It may help clarify the working philosophy for further discussion. 

Author Response

Reviewer 1:

The presented study aims to assess the prognostic value of 18F-FDG imaging on gastric cancer patients using a retrospective radiomics approach. This study is a side project to a nationwide multi-centered PLASTIC clinical study. The authors extracted over 100 features out for an overall comparison; however, none of the features showed significance when tested with the outcomes of patients. The conclusion is straightforward, following the results that the use of FDG PET imaging has a limited improvement of prognosis over radiomics analysis. 

Despite the overwhelming and clearly stated format, some critical questions remain unsolved in this study. 

Thank you for your comments regarding our manuscript. We appreciate your time and effort reviewing our article and your suggestions.

  • It is well-known that SPECT and PET are limited in their spatial resolution and anatomic information. The attenuation was calibrated by the CT while the image reconstruction to eliminate the noises. I don't find a clear statement whether the authors did radiomics analysis using both CT and PET information or, basically, PET. The only figure (Fig 1) suggested the VOI of the PET image, so it is anticipated that the radiomics was done using the PET signals only. Compared to CT, PET radiomics are relatively rare. The authors may want to have a more comprehensive discussion or, alternatively, include CT radiomics to suggest if PET or CT images are more feasible for radiomics approaches. 

Thank you for your feedback. Nuclear imaging techniques are indeed limited in their spatial information. However, they contain functional or molecular information. In [18]FDG PET specifically, tracer uptake (heterogeneity) might contain metabolic information about the tumour, for example by quantifying areas of (central) necrosis. With [18]FDG PET radiomics, it is aimed to quantify tracer uptake heterogeneity and other tumour characteristics. The field of PET radiomics is emerging with over 400 Pubmed publications in 2020 (PMID 34537132), compared to over 1000 today (11th May 2023).

In this manuscript, we only analysed the PET images and purposely left out the low-dose CT. In the PLASTIC trial, [18]FDG PET was compared to diagnostic CT. We are currently exploring whether handcrafted of deep learning radiomics could be exploited to analyse these diagnostic CT scans. Therefore, we chose not to include ldCT radiomics in this analysis. We added to the discussion line 340-343:

“Radiomics derived from contrast-enhanced CT has already shown promising results [13-15]. Since diagnostic CT is incorporated in the Dutch national guidelines of gastric carcinomas, future studies might focus on handcrafted or deep learning radiomic analysis of these scans.”

Also, it was clarified that radiomic features were extracted from PET features only in line 150:

“Tumours were delineated on [18F]FDG PET scans using an…”

  • Challenges for PET radiomics studies have been widely discussed in many articles. The authors may want to address or comment on the existing arguments, especially when their findings were negative to the use of radiomics prognosis. 

PET radiomics indeed comes with several challenges, the most important being the limited spatial resolution, the small datasets, and dependence on [18F]FDG-avidity.

However, PET radiomics comes also with some strengths, as PET, as no other imaging modality, allows for quantitative imaging. In addition, PET radiomics offers distinct advantages over CT radiomics due to its ability to provide functional imaging, capturing valuable metabolic information

The strengths and limitations section of the discussion has been extended (line 321-336):

“ Our study has several strengths and limitations. A strength is the relatively large number of patients. Most radiomic studies in nuclear medicine consider relatively small patient cohorts, as a result of the specialised nature of the imaging procedures compared to for instance CT. In addition, [18F]FDG-PET/CT scans were collected from sixteen health care institutes in the Netherlands, which is both a strength and a limitation. As a result of the multicentre setting, a larger cohort of patients could be obtained. In addi-tion, [18F]FDG-PET/CT-scans were acquired using different scanners and reconstruction protocols, which is representative for the clinical practice. However, the variation in imaging protocols was also challenging, since not all scans were EARL-compliant. This increased variability and reduced the repeatability and reproducibility of the extracted radiomic features [29]. To minimise the difference between EARL-compliant images and images reconstructed with clinical, site-specific reconstruction protocols, ComBat harmonisation towards the EARL-compliant scans has been performed [25]. Another limitation of the study was the VOI delineation. Twelve patients were excluded, because lesions could not be detected on [18F]FDG PET/CT. Low [18F]FDG-avidity also compli-cated the VOI delineation of some included lesions.”

  • It is suggested to include an overview or flow chart for the radiomics analysis using PET data. It may help clarify the working philosophy for further discussion. 

Thank you for the excellent suggestion. A flow chart was added to the methods.

Figure 1. Schematic overview of the statistical analysis including ComBat harmonisation of the features from non-EARL-compliant images to EARL-compliant-images, 100-times repeated ran-dom split, stratified for the presence of metastases (80% training, 20% test), standardisation (µ=0, σ=1), redundancy filtering of the Pearson correlation matrix (r=0.9), and fitting of a LASSO re-gression classifier.

Reviewer 2 Report

Dear authors,

I had the opptortunity to read your mansuscript. The findings are relevant, I think one major limitation (site-specific image recontruction as already mentioned in the section "limitations") should be discussed in more detail. Please include a paragraph in the section "methods" dealing with "ComBat harmonisation towards the EARL-compliant 317 scans has been performed [25]".

Kind regards

Author Response

Reviewer 2:

Dear authors,

I had the opptortunity to read your mansuscript. The findings are relevant, I think one major limitation (site-specific image recontruction as already mentioned in the section "limitations") should be discussed in more detail. Please include a paragraph in the section "methods" dealing with "ComBat harmonisation towards the EARL-compliant 317 scans has been performed [25]".

Kind regards

Thank you for your comments regarding our manuscript. We appreciate your time and effort reviewing our article and your suggestions.

The section on ComBat was extended (line 168-177):

“ComBat harmonisation was used to harmonise the features extracted from the non-EARL-compliant images to features of the EARL-compliant-images [24]. Since raw imaging data were not stored, it was not possible to harmonize images posteriori. Therefore, other ways to harmonise radiomic features were investigated. ComBat harmonisation is a method that originates from genomic research, and is able to compensate for batch effects introduced by e.g. scanner or protocol effects. ComBat was directly applied to features already extracted from the images. A Kolmogorov-Smirnov test was performed to test differences in the distribution of SUVmean between EARL-compliant and non-EARL-compliant images, and ComBat was applied in case of a significant difference in distributions [25].”

Reviewer 3 Report

This paper addresses the role of PET-CT, or rather different machine-learning models, in detecting distant and peritoneal metastases in gastric cancer. It uses data from 236 patients of a multicenter prospective trial to conclude that minimal, clinically non-relevant, improved performance does not outweigh the arduous radiomics effort.

This study is fluently written and of timely importance. The study question is interesting, and the methods represent the actual real-world situation. It certainly merits publication in this journal with only minor corrections.

I only have a few questions:

Will the ongoing evolution of AI have an influence on the PET-based detection of metastases or is the limit rather the method PET-CT in the context of gastric cancer itself?

Does the difference between the results of Xue and yours lie in the model with a different balancing of certain parameters or is it the dataset?

Author Response

Reviewer 3:

This paper addresses the role of PET-CT, or rather different machine-learning models, in detecting distant and peritoneal metastases in gastric cancer. It uses data from 236 patients of a multicenter prospective trial to conclude that minimal, clinically non-relevant, improved performance does not outweigh the arduous radiomics effort.

This study is fluently written and of timely importance. The study question is interesting, and the methods represent the actual real-world situation. It certainly merits publication in this journal with only minor corrections.

Thank you for your comments regarding our manuscript. We appreciate your time and effort reviewing our article and your suggestions.

I only have a few questions:

Will the ongoing evolution of AI have an influence on the PET-based detection of metastases or is the limit rather the method PET-CT in the context of gastric cancer itself?

This is a very interesting question with a speculative answer, involving many different factors. It is difficult to say how the field of AI will evolve. The capabilities of AI have definitely improved a lot in recent months/years, showing promising and ready-to-use solutions in everyday life. However, in medicine, the application of new tools generally takes more time. Also, it should also be noted that tools like ChatGPT are very prone to errors and caution should be taken utilizing tools like these. Errors like these should be ruled-out, especially before using these tools in clinical practice. An informative article can be found here https://link.springer.com/article/10.1007/s00259-023-06172-w .

Deep learning radiomics may eventually bring about new insights about tumour biology in gastric cancer. However, from a clinical perspective, [18F]FDG-PET/CT is not found useful for the evaluation of gastric carcinomas. Therefore, [18F]FDG-PET/CT will not be used in the clinical work-up and no more training data will be generated. The amount of training data that is currently available is not enough for training [18F]FDG-PET/CT deep learning AI models. Therefore, it may be worth exploring deep learning radiomics derived from diagnostic CT first, since these are acquired in the clinical work-up. This was referred to in line 340-343 of the discussion:

“Radiomics derived from contrast-enhanced CT has already shown promising results [13-15]. Since diagnostic CT is incorporated in the Dutch national guidelines of gastric carcinomas, future studies might focus on handcrafted or deep learning radiomic analysis of these scans.”

Does the difference between the results of Xue and yours lie in the model with a different balancing of certain parameters or is it the dataset?

The differences between the results of Xue et al. and ours lie in both factors. It is therefore difficult to directly compare the results. In the first place, Xue considered an Asian population, which is different in disease characteristics from our Dutch population. Xue considered peritoneal metastases only (31%), while our study considered both peritoneal and distant metastases (21%). The selected parameters of Xue also differed from ours. We tried to contact Xue et al. to externally validate their model in our population, but unfortunately, we did not get a response.

Please refer to discussion line 279-303:

“Similar to our study, this study by Xue et al. compared a clinical model, an [18F]FDG-PET based radiomic model, and a clinicoradiomic model to identify peritoneal metastases in gastric cancer, resulting in validation AUCs of 0.87, 0.69, and 0.90, respectively. It is challenging to directly compare these results to ours, since Xue et al only identified peritoneal metastases, while our work tried to identify peritoneal as well as distant metastases. However, a trend can be observed when comparing the results of Xue et al. to our subgroup analysis of the intestinal and mixed-type tumours. For both analyses, the clinical models performed better than the radiomic models and the clinical models were only slightly improved by the addition of radiomic features in the clinicoradiomic model. Therefore, the value of [18F]FDG-PET radiomics in both studies is limited, when these slight improvements in model performances are compared to the laborious ra-diomic analysis. In addtion, our clinical and clinicoradiomic models performed sub-stanstially worse than the models of Xue et al. This might have been caused by the imputation of our clinical variables. For three of the eight clinical variables, more than 20% of the values were missing. By excluding patients with missing data, almost 50% of the data would have been discarded. Therefore, missing variables were imputed based on fourteen other clinical and treatment variables. The imputation may have negatively impacted our clinical and clinicoradiomic models.

Furthermore, a class imbalance might have complicated the classification task. Our dataset consisted of 21% of patients with peritoneal and distant metastases, compared to 31% of patients with peritoneal metastases in the study of Xue et al. As a result, the machine learning algorithm was trained using data with an overrepresentation of the characteristics of the majority class, while underrepresenting the characteristics of the minority class, i.e., the presence of metastases.”